# Mandan, Hidatsa, and Arikara Nation Perspectives on Rez Dogs on the Fort Berthold Reservation in North Dakota, U.S.A.

**DOI:** 10.3390/ani13081422

**Published:** 2023-04-21

**Authors:** Alexandra Cardona, Sloane M. Hawes, Jeannine Cull, Katherine Connolly, Kaleigh M. O’Reilly, Liana R. Moss, Sarah M. Bexell, Michael Yellow Bird, Kevin N. Morris

**Affiliations:** 1Institute for Human-Animal Connection, Graduate School of Social Work, University of Denver, Denver, CO 80208, USA; 2Department of Social Work, University of Manitoba, Winnipeg, MB R3T 2N2, Canada

**Keywords:** indigenous knowledge, colonization, rez dogs, reservation dogs, free-roaming dogs, cultural responsiveness, companion animals, access to care, animal control

## Abstract

**Simple Summary:**

Free-roaming dogs, also referred to as reservation dogs or rez dogs, hold important and unique roles in Indigenous communities. The purpose of this study is to document the cultural significance of rez dogs, challenges related to rez dogs, and community-specific solutions to rez dog issues affecting community health and safety from the perspective of 14 members of the Mandan, Hidatsa, and Arikara (MHA) Nation, also referred to as the Three Affiliated Tribes (TAT), who live on the Fort Berthold reservation in North Dakota, U.S.A. The primary intervention areas described by the participants included: culturally relevant information sharing, improved animal control policies and practices, and improved access to veterinary care and other animal services.

**Abstract:**

The research on the relationships between free-roaming dogs, also referred to as reservation dogs or rez dogs, and Indigenous communities is extremely limited. This study aimed to document the cultural significance of rez dogs, challenges related to rez dogs, and community-specific solutions for rez dog issues affecting community health and safety from members of the Mandan, Hidatsa, and Arikara (MHA) Nation, also referred to as the Three Affiliated Tribes (TAT), who live on the Fort Berthold reservation in North Dakota, U.S.A. One hour semi-structured interviews with 14 community members of the MHA Nation were conducted in 2016. The interviews were analyzed via systematic and inductive coding using Gadamer’s hermeneutical phenomenology. The primary intervention areas described by the participants included: culturally relevant information sharing, improved animal control policies and practices, and improved access to veterinary care and other animal services.

## 1. Introduction

Domestic dogs (*Canis familiaris*) and humans have cohabitated for thousands of years, and dogs have been readily integrated as companions, guardians, and working partners in many societies around the world [1,2,3,4]. According to recent estimates, there are between 700 million and 1 billion dogs globally, and about 70–75% of dogs are considered free roaming [5,6,7]. The World Society for the Protection of Animals defines free-roaming dogs as “dogs that are in public areas and not currently under direct control” [8]. This definition is typically applied to owned dogs who are allowed to roam freely, dogs who were previously owned but have since become lost or abandoned, and dogs who may have never been owned [9]. Some of these free-roaming dogs live within and alongside Indigenous communities in North America and hold a unique and important status in this context. In an Indigenous context, free-roaming dogs can also be referred to as “rez dogs”, short for reservation dogs, because of the land on which they roam [10]. While the imposition of settler colonialism and the impact of modernization have evolved and changed traditional practices related to rez dogs in Indigenous communities, rez dogs continue to be an integral part of many Indigenous Peoples’ social environment, carrying deep cultural meaning, spiritual significance, and roles that extend further than companionship [2,11,12].

There is a critical need to address concerns regarding free-roaming dogs’ impact on community health and safety (e.g., dog bites, aggressive behavior, food competition or predation on native and endemic wildlife, livestock predation, and zoonotic disease) [2,12,13,14,15,16,17]. Current management strategies have been limited by their inability to address the complex social, cultural, and structural factors that have informed historical and present practices related to free-roaming dogs. For example, recent literature has supported increasing access to veterinary care as a strategy to improve the health and well being of free-roaming dogs [18]. However, efforts to increase access to veterinary care for Indigenous communities have been significantly restricted due to financial constraints, the remote or rural location of Indigenous communities, and the prioritization of other issues, such as “inadequate housing, water supply, and sanitation” [19]. In the face of this resource gap, non-profit animal sheltering and rescue organizations have been the primary providers of animal services in Indigenous communities around the globe [3,20]. Unfortunately, the literature indicates that this approach is ineffective, resource intensive, and unsustainable as a long-term strategy for managing rez dogs within Indigenous communities [21].

To improve the health and safety of their communities and the growing rez dog population, many Indigenous communities in North America have developed population management strategies specific to their communities [22]. Without other effective or sustainable alternatives, some communities have resorted to lethal strategies for management, including culling their free-roaming dog population. However, this practice has been negatively received in most communities [9]. Furthermore, while culling may quickly reduce the population of dogs, it cannot stabilize the population without also being paired with a fertility control program that can reduce the rate of new births [23]. Initial research indicates that population management strategies that incorporate culturally responsive communication and culturally specific education may be more effective and regenerative than culling or fertility control alone [11]. However, community-specific knowledge and perspectives on rez dog population management within Indigenous communities remain poorly documented in the scientific literature. 

This study aimed to document the traditional and contemporary beliefs, attitudes, and strategies for management and care of the rez dog population within the Mandan, Hidatsa, and Arikara (MHA) Nation, known together as the Three Affiliated Tribes (TAT), who are currently residing on the Fort Berthold reservation in North Dakota, U.S.A. [24]. Historically, the Fort Berthold reservation has been the home for a large and growing population of rez dogs with few laws or policies that address dog health and dog-related risks for community members [25]. This study originated from MHA Nation members identifying the need for a coordinated effort to address the issue of free-roaming dogs in their community. By documenting these perspectives, this study can be utilized to develop culturally appropriate and community-specific initiatives for the MHA Nation.

## 2. Materials and Methods

### 2.1. Design

Gadamer’s hermeneutic phenomenological approach was used to gain a better understanding of how the cultural significance of rez dogs within the MHA Nation could be incorporated into future free-roaming dog management strategies. Phenomenology is typically used when there is limited research on a topic and researchers want to gain a collective understanding of a phenomenon based on the lived experiences of a group of individuals [26,27]. Gadamer’s hermeneutic phenomenology uses a reciprocal interpretation process to further integrate the knowledge and interpretations of the researchers with the lived experiences of the participants to generate a nuanced description of the phenomenon. Due to the diversity of perspectives included in the analysis process, the findings of studies conducted using this method have cultural and historical accuracy as well as increased scientific credibility [26,27].

Gadamer’s hermeneutical phenomenology is based on three philosophical principles: recognizing preconceived notions, pursuing a “fusion of horizons”, and developing a “hermeneutic circle” [26,27]. Prior to engaging in the research process, the research team shared and reflected on their own lived experiences around rez dogs (“preconceived notions”). The research team members represent various disciplinary backgrounds (animal sheltering; social work and Indigenous studies; academic research; community organizing; and community outreach) which formed the basis of the preconceived notions that were brought into the study. The “fusion of horizons” principle of Gadamer’s hermeneutical phenomenology was operationalized in this study through the development of the interview questions and through the data analysis process (see Section 2.3 and Section 2.4). Finally, the “hermeneutic circle” principle was achieved through the peer debriefing sessions of the research team and through the presentation of direct quotes from the participants in this manuscript (see Section 2.5).

The researchers also evaluated the research procedures prior to data collection, striving to minimize bias and potential harm to participants who engaged in the study. Interview questions were evaluated for their capacity to recognize and validate the lived experiences of tribal members by acknowledging privilege and power dynamics in researcher–participant relationships. During consent administration, researchers informed participants of their right to decline to answer any questions and their right to cease participation in the study at any time.

### 2.2. Participant Recruitment 

This study was conducted in partnership with Indigenous community members and tribal leaders from the MHA Nation. The research team collaborated closely with Michael Yellow Bird, MSW, PhD, and a member of the MHA Nation, on most phases of this study, including participant recruitment and the preparation of this manuscript. Dr. Yellow Bird is a Professor and the Dean of the Faculty of Social Work at the University of Manitoba, a scholar of Indigenous studies, and a respected and integrated member of his Indigenous community. Dr. Yellow Bird directed the participant recruitment and provided the research team with a list of MHA Nation community members consisting of his personal and professional connections to contact for the research study.

Recruitment for the study was carried out under a University of Denver Institutional Review Board (IRB) protocol (DU IRB Protocol #767926). Eligibility for the study required participants to be over the age of 18 and members of the MHA Nation. Fourteen participants were selected to participate in the study. The tribal members recruited for this study were community elders, animal control officers, and general community members residing on the Fort Berthold reservation.

The demographics of the participants have been included in Table 1. Of the 14 participants, 6 (43%) self-identified their gender as women and 8 (57%) as men. In addition, all 14 participants (100%) self-identified as Native American, American Indian, or First Nations. Finally, 10 (72%) were 30–59 years old and 4 (28%) were 60 years or older.

### 2.3. Data Collection

Participants completed a one hour semi-structured interview in August 2016 with four members of the research team. The participants did not receive compensation for their participation. To facilitate the hermeneutic process, the questions for the interview were developed so that participants could reflect deeply and openly on their experiences with rez dogs [26,27,28]. First, the researchers asked participants about their experiences with rez dogs in the present. Second, participants were asked about the cultural significance of rez dogs. Third, the participants were asked to share their suggestions for how to improve the community relationship with and elevate the cultural significance of rez dogs. The interviews were conducted in the setting that was preferred by the participants (e.g., their personal homes, their place of work, or a public location). Interviews were audio recorded and de-identified by the research team. The de-identified audio files were then transcribed by a second-party vendor (GoTranscript, Ltd., Harrow Middlesex, United Kingdom) and stored in a password-protected database. To ensure the accuracy of the transcriptions, a member of the research team reviewed the transcripts while listening to the recording and made corrections if there were any inconsistencies between the transcripts and the recording. These transcripts were then archived prior to analysis due to staffing changes on the research team.

### 2.4. Data Analysis

In 2022, the archived transcripts were systematically and inductively coded for themes by five members of the research team. To begin, the researchers read each transcript to determine each individual’s overall experience (“immersion”) [26]. The second phase involved developing interpretive summaries of the themes that emerged during the interviews. A codebook was then developed based on the identified themes. The codebook was compiled based on the collective consensus among the researchers. A total of 11 primary codes (themes) and 32 subcodes (sub-themes) were included in the codebook. In the fourth phase, interviews were divided among the researchers, and the codebook was applied to the interviews using qualitative coding software (ATLAS.ti, Version 3.19.0; ATLAS.ti, GmbH, Dartmouth, NS, Canada). To reduce researcher bias, each interview was verified for coding accuracy by a second researcher after the initial application of the codebook. The coded data were then reviewed, and themes were organized and grouped according to their common meaning (“aggregation” and “illumination”) [26]. Throughout the analysis process, researchers developed an iterative understanding of the phenomenon by evaluating how they have come to experience as a result of listening to the participants’ lived experiences (“fusion of horizons”) [26,27,29].

Finally, to fulfill the final philosophical principle of the method, researchers continuously reflected on how their meanings connected to and differed from those of the participants (“the hermeneutic circle”) [26]. The research team then generated a summary of the participants’ shared experiences related to the phenomenon. The resulting list of themes and participant quotes most representative of the interview and the participant’s experience have been reported here. Participant interviews are cited at the end of each quote with “P” (participant) followed by the number associated with that participant’s interview. For example, Participant 4′s interview would be cited as “P4”.

### 2.5. Validity and Trustworthiness 

To improve the validity of the findings, the research team engaged in several peer debriefing sessions to critically assess their own interpretations of the data and to identify how assumptions and biases were informing their interpretations of the data [30]. As many direct quotes as possible have been included in the results to facilitate the reader’s validation of the findings [26].

## 3. Results

### 3.1. Cultural Significance of the Dogs

Throughout the interviews, the MHA Nation members who participated in the study shared the variety of ways in which rez dogs have current and historic significance in their culture. Participants described rez dogs as “sacred” and “part of our culture”. This cultural significance manifests in traditional oral stories passed down from generation to generation and in present-day interactions with dogs.

Throughout the interviews, participants discussed rez dogs’ role in cultural ceremonies and stories. One tribal member emphasized “We have origins that include the dog. There were songs for it” (P8). Another tribal member shared “I believe we used the dog in ceremony […] They revived some of our Sun Dances with the Mandan/Hidatsa/Arikara people” (P5). For the MHA Nation, dogs are a living symbol of courage, loyalty, perseverance, protection, and wisdom.

Moreover, participants detailed the reciprocal relationship between dogs and humans. They explained that dogs played several important roles, including serving as protection, guardians, pulling travois to transport materials, and as partners in hunting efforts. One participant highlighted these roles:


*“In the books on Maxhidiac or Waheenee, there are some stories about the dogs and they talk about the importance of the dogs to us here. Number one for protection, because many times … the old stories tell of people coming in to raid us. Other tribes would raid us. The dogs helped sound the alarm. The dogs will help in the fight when it occurred. But they were also useful to travois” (P8).*


One tribal member summarized how the cultural significance of dogs translates to an emphasis on the humane treatment of the dogs: “Well, the stories that we heard was, it was how we treated [dogs]. They say if they mistreat [a dog] and killed [a dog] that they will live a terrible life and death” (P6). Another tribal member noted, “Indians don’t like to kill dogs […] That’s why they are all over […] I tell you, what my Uncle’s telling me, they’re powerful. Dogs protect us from evil” (P9).

Participants also described the present-day relationship between people and dogs on the reservation as deeply interconnected. Participants noted, “You can gauge the health of the community by the health of their animals and vice versa,” (P1) and “When people suffer animals suffer too” (P1). Many participants described continuing the reciprocal relationship between dogs and humans in the present day by acknowledging their spiritual significance and by demonstrating care and respect for the dogs.

#### 3.1.1. Role of Dogs

Several participants stated that dogs continue to have a substantial role in the culture of the MHA Nation and they are considered a source of “protection” and/or “part of the family” (P4). Multiple participants referred to the dogs’ ability to anticipate danger: “They know when anything bad is going to happen. We should all be taking care of these animals because these animals are going to protect us” (P11). Another participant shared the belief that dogs “know when people are going to die. They’re messengers” (P11).

Another tribal member described the integral role of dogs in child care:


*I raised my brother’s kids. The dogs were my babysitters. Through the years, I think, I’ve had maybe 4 or 5 dogs, and when I was in the house cooking or doing whatever, if I went outside and I look for the kids and I didn’t see them, I’d whistle for my dog and whatever house she came around, I know where the kids were because that’s where they were. She was my babysitter and she did take really good care of them (P6).*


Participants also noted the role of dogs in ceremonies and traditional stories. One participant tells the story of a village surrounded by soldiers that was led to safety by a female dog and her litter by going under a river (P11). Another traditional story detailed nations of cats and dogs pitted against each other in a generational conflict in which the cats stole the tails of the dogs. This traditional story was used to explain the origin of dog greeting rituals by saying “It looks like they’re smelling each other’s you-know-whats, but what they are really doing is looking for their tail that they lost” (P1).

#### 3.1.2. Colonization and Cultural Disruption

Many tribal members detailed the effects that colonization and cultural disruption had on the community’s relationship with dogs, describing colonization as a “cultural clash” that causes a “disconnect” and that it meant that “we got so distanced from who we were” (P10). One participant summarized, “It’s understood that a lot of our culture is not the way it used to be” (P11).

Several participants spoke about the implications of colonization on tribal members’ relationships with the dogs: “You’re talking about serious culture disruption. Animals, dogs, people, culture. All that sort of stuff, relationships break down” (P14). Additionally, participants stated “The dogs got set aside,” (P14) and “People used to be more connected to take care of the dogs” (P8).

### 3.2. Challenges Related to Free-Roaming Dogs

The interview participants had various perspectives on the extent to which rez dogs represented a challenge for their community. The majority of the participants agreed that dogs were roaming in their community. However, there was disagreement on how many there are and if they pose a negative issue for the community. While some tribal members felt strongly that the dogs were an issue for their community, others felt that the dogs were not an issue at all. Some participants reported that there are fewer roaming dogs than there used to be, while for others, it felt like the number of dogs was the same or increasing. Most tribal members noted that the number of dogs and severity of issues varied by location on the reservation. 

Community members listed several factors that contribute to the challenges related to free-roaming dogs on the reservation. Many tribal members cited growth in the human population and an overwhelmed animal control agency as possible reasons for the number of free-roaming dogs on the reservation. One tribal member felt that poverty was the “main link”, explaining that the areas with a high prevalence of roaming dogs were also “the areas that tend to have more economic difficulties” (P13). Meanwhile, other community members identified “irresponsible pet ownership” and a small number of “repeat offenders” who choose not to spay/neuter their dogs and/or let their dogs run loose as the cause for the number of free-roaming dogs (P7). Moreover, some tribal members felt that community members might be abandoning puppies once they get older, saying, “they think they are cute when they are tiny, but after they get older, then they just don’t care about them” (P4). 

Many participants shared a concern that the free-roaming dog population was growing from sources outside the reservation, such as oilfield workers, casino patrons, or other non-enrolled members who abandon their dogs on tribal land. One participant shared their ideas about where the free-roaming dog population originates:


*[The free-roaming dogs] live in the abandoned houses and trailers, old vehicles, just pretty much anywhere that they can seek shelter. That is when they start gathering in packs and start getting that pack mentality because they’re trying to stay warm, they’re trying to survive that cold winter (P3).*


Furthermore, there is a lack of consensus on how or if dogs should be contained when they have a designated family or owner. One tribal member explained: 


*[I] heard it stated that some families don’t think that animals should be contained. They should have the ability to roam and the freedom to go about their business. They don’t think that they should contain them. And whether that’s just personal views or social views, I don’t know, it’s hard to say (P13).*


Lastly, one participant discussed how the condition of the kenneling facility on the reservation serves as a barrier to housing dogs:


*One of our major limiting factors is our kennel*
*—the condition of our kennel. The director before me had this kennel built out of Quonset, and it’s by no means adequate to be a kennel, but we made do with what they have (P7).*


#### Animal Control

Several participants highlighted their desire for improved animal control policies that could address the community’s health and safety concerns. In particular, participants shared their worry about the free-roaming dogs’ potential to harm humans as a key issue to address. Some tribal members likened the rez dogs to “wolf packs who attack people as well as other dogs and animals, but back off if you stand your ground” (P2). Another participant said that the rez dogs were “constantly running on the road” (P11), while others described the rez dogs as “hungry” and said “they devour everything” (P12). Many tribal members characterized the rez dogs as “aggressive” (P8), sharing that some dogs bark at people or chase them which prevents community members from being able to walk outside without fear. One participant shared: 


*I used to be an avid outside walker. We’ve since purchased an indoor treadmill because depending on the breed and the size, the dogs are dangerous. If they don’t know you, they may not be that nice. I’ve had family members bitten, chased where they had to jump on top of something. But as far as recreational activities being able to walk and having that freedom of your community, I see the dogs as being difficult (P5).*


Participants discussed how the breeds of the rez dogs in the community have shifted over time and attributed these changes to the rising safety concerns. One participant shared: “I don’t think the rez dogs were as bad growing up” (P12). Another tribal member elaborated that some breeds, such as pit bull type dogs and Rottweilers “are not necessarily what I consider the animal you should have in my town” (P10). Another member echoed distrust of certain breeds because dogs, such as German Shepherds were “attacking people and hurting them” and that they are “not Indian dogs” (P9). As a result of these safety concerns connected to specific breeds of dog, animal control officers who participated in the study noted that in some areas of the reservation, breeds, such as Rottweilers, pit bull-type dogs, and Mastiffs, were banned by laws or ordinances.

Conversely, other tribal members categorized the behavior of free-roaming dogs more positively. Animal control officers who participated in the study estimated that they see only 10–20 cases of dog bites per year and that they had come across rez dogs that “were actually well trained” (P7).

In addition to safety concerns about dog bites and attacks, tribal members noted a few additional concerns related to the rez dogs. Diseases, such as canine brucellosis, mange, parvovirus, and flea/tick-borne diseases, were all listed concerns regarding free-roaming dogs. Notably, the participants did not describe rabies as an issue in their community.

Participants described several limitations of existing animal control policies to address the free-roaming dog challenges on the reservation. Several participants noted that while there is a leash law in place that should limit the number of free-roaming dogs, that law is not well known by community members. One participant described the need to improve public knowledge of the leash laws:


*[Animal control officers] try to educate them [when they come to pick up their dog] because education is the most important thing. A lot of people claim that they had no idea that there was a Leash Law, so whenever they come in to get their dog, [animal control officers] explain to them the reason why we picked up your dog; it was at-large and then, here’s the fees for kenneling and everything. You have a certain amount of days to come and retrieve your dog. [Animal control officers] try to educate them as best as we can (P7).*


Other participants were opposed to laws that would restrict the free movement of dogs. One participant discussed the historic role of dogs for the MHA Nation, the concept of ownership traditionally, and how these ideas “clash” with the policies that have been put in place:


*Well you know that’s kind of how people lived back then. They didn’t have fences up. They didn’t understand that*
*—hey there’s a restriction here*
*—horses roam, dogs roam. But this law is saying now you live in a different world that says you can’t do that any longer. Right? And so it’s like a cultural clash (P14).*


Interview participants shared that their most desired outcome for free-roaming dogs is that those dogs are returned to their owner or to the community. One participant described informal approaches for getting the dogs back to their owners that did not require getting animal control involved:


*[When we pick up a dog] the local tribal newspapers started putting them in the paper and just word of mouth, we get it on the tribal email system, and people say, ‘Oh that’s someone else’s dog’. So they let them know (P7).*


Participants detailed other approaches for managing free-roaming dogs. One participant noted


*“Not everybody thinks that they are sacred people roaming the earth. There are people who say ‘the heck with that idea.’ They want them dead” (P12).*


A central theme of many interviews was the challenges that animal control experiences in the community. One interviewee said:


*I think it’s frustrating for the animal control workers. They come in very excited to help. They want to do their best. They want to help animals. They just get burned out and run down by this constant source of picking up these dogs (P13).*


A few animal control officers discussed the difficulties of the job: “You not only have enrolled tribal numbers, but you also have non-members too, so it’s hard to enforce that over the jurisdiction of non-members” (P7). They emphasized one of the problems in the community is people who repeatedly leave their dogs out to roam freely: “We spend so much time going after repeat offenders, and we don’t get time to reach out to the community very often” (P7).

Another challenge the officers shared was the stigma faced by animal control in the community: 


*“The huge misunderstanding or misconception within our community on the reservation that animal control is perceived as dog catchers, or dog killers even. They think that once [animal control] gets your dog, they shoot the dog” (P7).*


### 3.3. Community-Specific Solutions

#### 3.3.1. Culturally Relevant Information Sharing

When reflecting on how to best address the current challenges related to free-roaming dogs, many participants identified a need for culturally relevant information and education on how to care for the rez dogs. Many participants suggested that this education focuses on young people. One tribal member summarized the most important topics for these information-sharing efforts: “Number one they need to know how important these animals are and that we’re here to take care of them. They’re given to us for a reason” (P11).

Given the identification of the behavior of some of the dogs being one of the primary concerns, one tribal member suggested offering animal handling and training classes for the community:


*I’d like to see classes on training animals and showing young kids. They’re going to see somebody show them how to respect and how to get certain things from an animal. They will just eat it up and love it and show off with it. And be proud of their animals (P11).*


Another participant suggested these information-sharing interventions occur in schools and believed those lessons could take a story format: “We could have like dogs tales. Where we tell old stories or something, those old stories since I have been here” (P14).

Another participant felt that information-sharing efforts should not just focus on young people and, instead, focus on the family unit as a whole: “It has to be family for me. Teaching begins from the family. Education [in schools] has played a big role, but for me, anything I learned in my culture starts at home” (P5).

Tribal members shared that information sharing on the cultural significance of rez dogs would not only help the community with caring for the free-roaming dogs but also work to combat cultural disruption and erasure caused by colonization: 


*I think that a lot of cultures are going extinct and we hear about that a lot. It would be fun to get a beautiful artist to do the drawings, and some of the elders to tell the stories and we write them down and publish them. And do it in different languages too. That is one thing I see missing, in children books in the two languages. It’s something you do with your mom and dad, and it’s kind of interactive. You have discussions about it (P14).*


#### 3.3.2. Community Care and Responsibility

The participants expressed varying perspectives about caring for dogs in their homes and community. When describing the general approach to taking care of dogs on the reservation, study participants expressed a common theme: “You gotta take care of them, the dogs have to be maintained. They have to be loved (P9)”.

Multiple study participants described community care and ownership of dogs on the reservation. Community care was provided most often by providing food and water for free-roaming dogs. Participants shared that “there are a lot of people willing to take in animals”, and “We don’t have any [dogs], but they come to our house all the time. We feed them. We have bags of dog food. Feed and water them” (P11).

According to the participants, some individuals have even taken on the responsibility of providing veterinary care and facilitating the adoption of dogs. One tribal member described how their daughter has “out of her own pocket has made sure they had their shots, were fixed, and then she doesn’t give them to just anybody either, or she makes sure they’re going to good people. There are so many people out there like that in our community” (P11).

#### 3.3.3. Access to Veterinary Care and Other Animal Services

Access to care was an essential theme for many of the participants. Nearly all participants emphasized the importance of improved access to veterinary care and animal services in improving the health and well being of rez dogs on the Fort Berthold reservation (P13). Multiple interviewees discussed the relationship between the free-roaming dogs on their reservation and the gaps they experience in access to veterinary care and other pet support services, including spay/neuter and basic veterinary care. When asked why many of the dogs on the reservation are not spayed or neutered, one respondent answered “access and money” (P11). A general theme expressed by participants was that “it takes a lot to take care of a dog, and you have to make that commitment” (P2).

Mobile veterinary clinics and/or private practice clinics located in adjacent cities are the primary forms of veterinary care for tribal community members’ animals. One interview participant highlighted the overall need to increase access to low-cost spay/neuter, vaccinations, and other veterinary care: 


*[A barrier] is just the resources that might be needed in the community for vaccination and for a spay/neuter and all those things that are needed to keep the dogs healthy. It’s, you know, one thing to educate people, but then you have to back it up with the resources so that we can provide the right level of care (P2).*


Participants shared that mobile veterinary clinics come to the community intermittently and offer free services. However, “it takes a lot for the owner to take [dogs] down to these little vet clinics” (P3), and “some people can’t get to them” (P8). Another individual indicated that the veterinary services are inconsistent and unpredictable: “This year, I think, is the first year they didn’t come up this way […] I wanted to get my dogs a few shots, but they didn’t come around this year” (P5).

Similarly, the participants expressed a lack of capacity for veterinary services to serve the community. One participant described one of the nearby veterinarian’s capacity as a limiting factor:


*[He] comes down and spends a day in Parshall or New Town and he spays and neuters out of his mobile clinic. He is only one surgeon though versus with that mobile clinic that could get thirty or forty done in a day. He can do five or six (P13).*


Even with the services provided by the existing clinics, community members expressed frustration with other barriers to accessing the services, such as scheduling processes and appointment restrictions. One community member recalled, “It’s on a first-come, first-serve basis. So if you’re not there right away at 5 o’clock in the morning signing up, you probably aren’t going to get seen” (P3). Another tribal member summarized their understanding of the communities’ experience attempting to access the traveling clinics:


*A lot of people had to call ahead of time to say they were bringing [in an animal]. And there were a lot of people that got turned away. Because they were hearing “just bring them in” and they were going to spay, neuter, whatever [… ] now they are making everybody do appointments (P11).*


Another participant described a program in the area that provides free vaccinations for animals adopted through the local animal control. The participant explained that being able to provide low-cost vaccinations was important. However, the current voucher-based system was ineffective and underutilized:


*We do a voucher system with animal control here for vaccinations. So, for an animal adopted out of animal control, they get a voucher saying that animal control will pay for the vaccinations. They take the voucher, bring the voucher down to [the veterinarian] and the vaccination is free. [The veterinarian] sends the voucher back to animal control. They reimburse [the veterinarian] for the vaccination. But [the veterinarian has] only seen two vouchers in six months (P13).*


Participants frequently shared that tribal community members were open to receiving veterinary care services if they could access them. One interviewee shared the high demand for mobile veterinary services:


*Judging by our waiting list and even now we are out of funding for those mobile vet services, but people stalk our office non-stop. A majority of our calls at the Game and Fish Office are animal control calls, because they want to know when the next clinic is. So, people want to do it (P14).*


One participant also highlighted challenges related to the lack of access to behavior training support in their community:


*It takes a lot of effort to get up to Minot for dog training classes. It’s a day because you have to drive up there, do your stuff, and drive back. It takes a lot of effort to do that (P13).*


Tribal community members noted several concerns around neglect of the free-roaming dogs. Some participants described issues around the lack of adequate food, shelter, and water for dogs or containing dogs using a tether or in a yard. Some described instances of dogs passing away “because nobody was feeding it. Nobody took that responsibility” (P8) or that certain dogs they personally cared for did “belong to somebody else, but evidently they are not taking care of them” (P4).

## 4. Discussion

This study identified cultural perspectives, traditions, and current or proposed strategies for the management of rez dogs living on the Fort Berthold reservation in North Dakota, U.S.A. By centering the perspectives and proposed interventions of Indigenous community members, this study represents an important step toward developing culturally responsive and community-specific strategies for addressing rez dog-related issues on the Fort Berthold reservation. The primary intervention areas described by the participants included: culturally relevant information sharing, improved animal control policies and practices, and improved access to veterinary care and other animal services.

### 4.1. The Need for Culturally Responsive Information Sharing

Similar to the findings of the present study, previous studies on the cultural significance of dogs in Indigenous communities have found that the rez dogs of the MHA Nation are viewed as guardians and companions, they have spiritual and cultural significance, and rez dogs are accepted and celebrated in society [2,3,11]. For many members of the MHA Nation who participated in this study, rez dogs are more than companion animals; they are spiritual allies and protectors of the land and people.

Participants shared how colonization has impacted the flow of cultural knowledge from tribal elders to the younger generations and, therefore, impacted the younger generations’ relationships with dogs. In this study, participants shared how interventions that support decolonization are critical to restoring both the traditional role of dogs and the dogs’ sovereignty within animal welfare practices and policies. Decolonization is defined as “the process of reversing colonization and the damage done by reclaiming Indigenous Knowledge and implementing that knowledge in the lands and communities of Indigenous Peoples physically, psychologically, and emotionally” [31]. Decolonization in the animal welfare context should center Indigenous Knowledge (the knowledge of Indigenous and local communities about their own interaction with ecosystems) in the development of animal welfare policies and programs in Indigenous communities [32]. For example, strategies recommended by participants included building free-standing shelters and feeding stations for free-roaming dogs which would allow the community to preserve and optimize their tradition of community care and management of the free-roaming dog population. Research on decolonization has also identified additional strategies that could support the health and well being of both Indigenous communities and rez dogs, including broader interventions like recognized sovereignty, self-determination, and reparations [3]. Future research could also investigate how lateral violence has influenced free-roaming dog management practices [33].

By demonstrating cultural responsiveness and centering the community-specific needs in their work with Indigenous communities, animal welfare service providers can establish more effective and sustainable programs and policies [34]. Developing programs that utilize culturally responsive communication and provide accurate culturally specific information when engaging with Indigenous communities requires an understanding of the dynamic nature of Indigenous Knowledge and how it should constitute the foundation of community-specific health and regenerative practices [35,36]. This knowledge is grounded in culturally distinctive concepts and diverse notions of health, well being, and resilience that bridge relationships among people, other animals, and the environment [37]. Indigenous Peoples have a rich history of diverse languages, strategies, and values in areas of identity, the land, and responsibilities of looking and taking care of each other that are tied to their traditional territory [38]. Therefore, animal service providers should not extrapolate community-specific programs to other communities. By engaging in their own relationship-building with their focus communities to understand the community-specific strengths, priorities, and needs, service providers can support the implementation of community-led and culturally relevant programs. 

For the MHA Nation, many participants prioritized the need for culturally relevant information sharing for the young people in their community. The participants were excited about the potential solution of building a center for humane education that could also function as an animal shelter and resource center. This “MHA Nation Center for Humane Education” would also be a place where the community, particularly young people, could engage in dog training classes. Several participants shared how having a physical building dedicated toward cultural revitalization and preserving the cultural significance of rez dogs would greatly enhance the community’s efforts to care for the dogs.

### 4.2. The Need for Community-Based Animal Control

The role of animal control/field services is to “provide a wide array of services to their communities, including saving pets in danger, protecting human health and safety, enforcing laws and ordinances, providing support and education to community members, disaster response, helping lost pets get home, and helping wildlife, livestock, and exotic animals, in addition to cats, dogs, and other pets” [39]. Free-roaming dogs fall within the purview of animal control/field services due to the ongoing discussion in the veterinary epidemiology and public health fields about the potential risk of free-roaming dogs to community health [14,15,16,17]. This discussion includes their negative impacts on native and endemic wildlife, they are common vectors for disease transmission (rabies, parvovirus, parasites, and canine distemper virus), and can be perpetrators of aggression towards humans [12,13].

However, research on the relationship between free-roaming dogs and various public health and safety concerns has produced mixed results. For example, studies conducted in urban/non-reservation environments have documented people being more likely to be attacked when approaching dogs that are chained up, fenced, or owned [40,41,42]. In Bali, free-roaming dogs were noticeably calmer, less likely to attack or chase other animals, and less excitable or active than owned/kept dogs [43]. Further, there are clear benefits to the dogs themselves in allowing them to roam freely [44,45,46,47,48]. In a study on animal sheltering in India, unrestricted movement (the capability to move freely about a neighborhood regardless of human control) was identified as the most important factor in upholding good welfare for free-roaming dogs [45]. Similarly, ethnographic interviews with the Indigenous Tofa communities of southern Siberia highlighted that dogs need the freedom to roam in order to live their fullest lives [46]. Future research should focus on developing a more complete understanding of how free-roaming dogs can be effectively managed so that equitable health for all components of a communities’ ecosystem is respected [3,7,45,47,49,50].

Due to the multi-faceted and community-specific nature of rez dog-related challenges, community engagement represents a promising approach to improving the effectiveness and sustainability of animal control/field services policies and practices on the Fort Berthold reservation [49,50]. A limited number of studies have discussed how community engagement strategies have been used in the animal welfare field [51,52,53,54]. Community engagement provides community members the opportunity to collaboratively identify issues and offer critical perspectives on the policies and practices that directly affect them. The approach can also identify individual and collective strengths, challenges, assets, and lived experiences that can be leveraged to improve the effectiveness and sustainability of policies and programs [55,56]. Findings from this study could be used to form the foundation of future Fort Berthold animal control officers’ community engagement efforts. For example, most participants in this study identified a need for improved information-sharing efforts around the current animal control policies of the Fort Berthold reservation, particularly the leash laws and shelter requirements for dogs who live outdoors. Community engagement approaches could facilitate animal control officers in identifying culturally relevant and community-specific approaches for information sharing that would be most effective at improving the health and well being of both the people and the rez dogs, while also gathering input on strengths or limitations of the current animal control policies.

### 4.3. The Need for Access to Veterinary Care and Other Animal Services

The majority of participants explained that they either participated in or witnessed the community care of dogs on the Fort Berthold reservation. This community care typically took the form of providing food and water to the dogs. Many MHA Nation members were known to connect rez dogs with veterinary care and permanent families and homes. These community care practices of feeding and sheltering are similar to those documented through research conducted in India, where “low barrier” forms of community care (i.e., feeding) are commonly practiced, with a few community members engaging in acts for long-term welfare (e.g., vaccinations, spay/neuter, shelter, etc.) [44]. These community care efforts represent a critical source of mutual aid in the community, which could reduce the burden on the already under resourced and overburdened animal control system on the Fort Berthold reservation. However, the MHA Nation members noted several barriers to engaging in these ongoing community care efforts, including insufficient access to veterinary care and other pet support services.

Access to veterinary care and other pet support services (e.g., grooming, behavior training, and pet supplies) was highlighted by most of the study participants as a challenge related to caring for the rez dog population on the Fort Berthold reservation. Research on barriers to accessing veterinary care is extensive. Research conducted in non-Indigenous communities throughout the U.S. has identified factors that exacerbate barriers to accessing pet support services, including affordability of services, geographic proximity to pet care services, transportation barriers, clinic hours of operation, bilingual services providers, positive and trusting service provider–client relationships and communication, client values concerning the importance of veterinary and other pet support services, and general access to pet care information [57,58,59,60,61]. Of these barriers identified across the U.S., geographic location of services, clinic hours of operation, and scheduling processes were identified as the most significant barriers to accessing services for MHA Nation members. 

Only a few studies have documented access to veterinary care as a barrier to community-wide health in Indigenous communities [3,10,62]. Many of the participants in this study highlighted a need for animal welfare service providers to be responsive to community-specific needs around how they structure their programming. For example, the participants described that while the mobile clinics were a valued source of veterinary care, they were often limited in their effectiveness. The participants discussed challenges related to the mobile veterinary clinics, as they were inconsistently scheduled and did not meet the demand for services on the reservation. To this point, there is an important distinction in the human health care literature between “having access” to services, meaning the potential to access a particular service, and an “gaining access,” referring to actual utilization of the service [63]. There are a variety of strategies currently being used in the animal welfare field to attempt to support remote communities’ moving toward actually gaining access to veterinary care, including providing telehealth options for non-surgical procedures, offering transportation for pets and their owners to and from appointments, providing pet support care and supplies alongside existing human service providers or events (e.g., food and supply pantries), and delivering pet supplies (e.g., food, treats, litterboxes, collars, and leashes) directly to clients’ homes at no cost [64]. Animal service providers working to serve the MHA Nation community members should consider what the participants have shared regarding the barriers they are experiencing to accessing the limited pet support services in their community and make efforts to facilitate the MHA Nation community in gaining access to their services. 

Finally, it is important to highlight how most animal control policies and programs in the U.S. are heavily influenced by the highly subjective definition of “responsible pet (companion animal) ownership” [54]. Unfortunately, some of the animal welfare literature has equated the strength of an individual or a community’s bond to animals to colonial and classist standards, such as their willingness to pay for services or allow the animal to sleep in their bedroom [65,66,67,68]. This narrative has reinforced implicit and explicit bias against socially and economically marginalized populations in the U.S., particularly Black, Indigenous, and People of Color (BIPOC) and communities experiencing poverty. These biases justify the assertion that companion animal ownership/caregiving is—or should be—only available for individuals who can afford all facets of companion animal ownership under any circumstances and who agree with and operate within the settler colonial definition of responsible pet ownership [69,70]. Fortunately, more rigorous studies have demonstrated that when structural barriers to accessing pet care services were addressed through a community-level intervention the pet owners’ race and ethnicity was not predictive of seeking pet support services [71]. Future research should aim to develop a more inclusive and equitable definition of concepts such as “the human–animal bond” and “responsible pet ownership” that can then be used to assess the effectiveness of various animal welfare interventions in promoting the health and well being of people and companion animals alike.

### 4.4. Limitations

This study has several limitations to consider. The study had a small sample size of 14 MHA Nation community members, with all participants over the age of 30 years, which should be considered when evaluating its representativeness for the entire MHA Nation. Future research could explore variations in perspectives with greater representation of the young people in the community. The use of phenomenological methods in this study was intended to gather an in-depth foundational understanding of free-roaming dog issues on the Fort Berthold reservation, and future research could utilize quantitative approaches to begin to generate evidence that would support concepts of generalizability. Additionally, Indigeneity is not a monolithic cultural identity, and all tribal communities have distinct cultural values and traditions. Only members from the MHA Nation were included in this study, which impacts the generalizability of the study’s findings to other Indigenous communities.

It is also important to note that the data were collected seven years before the analysis took place and that the data analysis in this study relied on archived data. While valuable, archived data may be less relevant as these perspectives may have changed since 2016, and the gap in time may make member checking less relevant [72]. The study’s findings would be strengthened with a follow-up study on current perspectives and needs regarding interventions. This study also highlights an opportunity and need for community-driven and participatory action research that is conducted through an ongoing partnership with the focus population.

Finally, the positionality of the research team should be noted when considering these findings. While one of the senior members of the research team (Dr. Michael Yellow Bird), who led the design and recruitment for this study, is a member of the MHA Nation, there was a lack of positionality within the data analysis team, as the majority are not Indigenous, and none identify as members of the MHA Nation. Three of the authors who conducted data analysis for this study identify as white women, one identifies as a Latina woman, and one identifies as a Denesuline Indigenous woman. There is an opportunity for future studies to explore similar issues and questions but with increased Indigenous representation in the positionality of the authors.

## 5. Conclusions

Future efforts to develop or evaluate community-specific free-roaming dog management strategies should integrate the community’s historical and contemporary understandings of rez dogs. The effectiveness and sustainability of the interventions will be improved by prioritizing cultural responsiveness and promoting capacity for community caregiving and increasing access to pet support services.

## Figures and Tables

**Table 1 animals-13-01422-t001:** Demographic characteristics of the interview participants (n = 14).

Participant Demographics		Frequency	Percentage
Gender	Woman	6	43%
	Man	8	57%
	Other (Transgender, Nonbinary, Two-Spirits, etc.)	0	0%
Race/Ethnicity	Native American, American Indian, First Nations	14	100%
	Asian	0	0%
	Black	0	0%
	Hispanic/Latino	0	0%
	White	0	0%
	Two or More Races	0	0%
Age	18–29 years	0	0%
	30–59 years	10	72%
	60+ years	4	28%

## Data Availability

Not Applicable.

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
