# Peer review of "Mandan, Hidatsa, and Arikara Nation Perspectives on Rez Dogs on the Fort Berthold Reservation in North Dakota, U.S.A."

_animals, 2023, doi:10.3390/ani13081422_

Round 1

Reviewer 1 Report

Have only one correction. The especies scientific name must be write in italic fonts. 

  (Specific comments)

1. What is the main question addressed by the research?

- 1. The relationship between humans and dogs on Indian reservations in the US.

2. Do you consider the topic original or relevant in the field? Does it
address a specific gap in the field?

- 2. The theme is original and relevant, as it demonstrates this relationship in a clear and relatively detailed way.

3. What does it add to the subject area compared with other published
material?

 - 3. Provides a profile of the relationship between people and semi-domiciled dogs on the North American Indian reservation. Which will probably be very different from the relationship between dogs and Brazilian indigenous people or African natives, for example.

4. What specific improvements should the authors consider regarding the
methodology? What further controls should be considered?

- 4. The article's approach is qualitative, therefore, I did not observe any methodological contribution to suggest.

5. Are the conclusions consistent with the evidence and arguments presented and do they address the main question posed?

- 5. Considering the qualitative approach, yes.

6. Are the references appropriate?

- 6. Yes.

7. Please include any additional comments on the tables and figures.

- 7. No additional comments.

Author Response

Thank you for reviewing our manuscript. We have italicized the scientific species name in the introduction.

Reviewer 2 Report

This is a very important discussion about how people from indigenous cultures relate to companion and commensal animals. Thus it certainly has merit to publish in the context of human-companion animal relationship

I would suggest that the authors strengthen their presentation in two areas:

a) following the rather high-flying references to Prof Gadamer's hermeneutic in the M&M section, the discussion is rather pragmatic and no furtther use is made of the scholarly method. I suggest to include some more discuision of philosophical/cultural aspects.

In the context in M&M I would also suggest that the 2nd and 3rd step of Gadamer's methodology is also already explained somehow near line 103 ff, not only much later in lines 168 ff (or at least refer to line 168 ff already near line 103)

My second suggestion ist to include some more literature on the biology, human-dog relationship etc from other field studies of feral/free-ranging/village dogs, a good summary e.g. can be found in the book by Range & Marshall-Pescini 2022).

Specifically, Corrieri et al on differences in behaviour between free roaming village dogs and owned/kept dogs in Bali could be interesting, because the interviewees in Cardona et al's study also refer to differences

regarding to the behaviour described in l 303 ff, please have a look at Lord et al 2009 on mobbing behaviour...

in line 314 - 321, again Corrieri et al come to mind...kept/owned dogs showing more unwanted behaviour than real village dogs

Author Response

1) Following the rather high-flying references to Prof Gadamer's hermeneutic in the M&M section, the discussion is rather pragmatic and no furtther use is made of the scholarly method. I suggest to include some more discuision of philosophical/cultural aspects.In the context in M&M I would also suggest that the 2nd and 3rd step of Gadamer's methodology is also already explained somehow near line 103 ff, not only much later in lines 168 ff (or at least refer to line 168 ff already near line 103)

RESPONSE: Thank you for the input on clarifying our methods section. We added a summary of how the 2nd and 3rd principles of the method were applied to Lines 110-115.

2) My second suggestion ist to include some more literature on the biology, human-dog relationship etc from other field studies of feral/free-ranging/village dogs, a good summary e.g. can be found in the book by Range & Marshall-Pescini 2022). Specifically, Corrieri et al on differences in behaviour between free roaming village dogs and owned/kept dogs in Bali could be interesting, because the interviewees in Cardona et al's study also refer to differences

regarding to the behaviour described in l 303 ff, please have a look at Lord et al 2009 on mobbing behaviour…In line 314 - 321, again Corrieri et al come to mind...kept/owned dogs showing more unwanted behaviour than real village dogs

RESPONSE: We appreciate the additional reference suggestions. We have integrated Lord et al 2009 and Corrieri et al, 2018 into lines 575-577.

Reviewer 3 Report

Wow! I am blown away by the quality of this submission. I am incredibly grateful that your research group explored this very important topic and I personally learned a great deal from reviewing this paper. Your introduction did an excellent job introducing the reader to the relevant material to understand your topic. Your methodology is appropriate for the study, and I am incredibly happy to see the amount of intentional effort that your research group took towards addressing bias in this study - an important aspect of qualitative research. 

My concerns mainly focus on confidentiality. As this is a qualitative study, I find the number of interviews appropriate with the focus of gaining a depth of knowledge rather than generalizability. However, there were 14 participants. In my experience with qualitative and quantitative studies working with human participants, the general rule of thumb is to not identify a subgroup with 3 or less individuals as these participants could be re-identified. 

LN 129-132: I think it’s interesting to know the different roles of the participants within the tribal community. However, I am concerned that the phrasing of this sentence could re-identify individuals, specifically the animal control officers, journalist and school principal. I would suggest revising this to protect the individual participants identities.

I would also suggest removing Table 1 (I know, I usually love a demographics table - this is a weird suggestion from me!) but I think you could just briefly describe the participants breakdown for gender and age and include a statement that all participants self-identified as Native American, American Indian and/or First Nations. 

Methods question - Was any compensation or direct / indirect benefits provided to participants? I would suggest adding a statement that outlines that in the Methods section. 

With regards to the Results and how statements were presented, I had a few more confidentiality concerns mostly related to how quotes were presented from participants that were identified by their role in the community, as it seems that they could be easily re-identified, and this could pose a risk to their reputation within their community. These quotes are all impactful and I believe that they should stay in the manuscript; however, I would suggest revising the presentation of the following lines to ensure that participant confidentiality is protected:

LN 266-269

LN 287-291

LN 314-317 * This may not need changed if you had > 3 animal control officers who participated.

LN 330-337 * This may be OK again if you had > 3 animal control officers who participated... However, may suggest removing the participant identifiers in general, ex. P7 as its clear P7 is an animal control officer.

LN 362-367

LN 461-477 * NOTE: I see how impactful these statements are because they come from the veterinarian and the member who works for the MHA Nation's Fish and Wildlife Division so they have that professional role; however, this would make each of these individuals easily re-identified.

Also, I understand from the article that the animal control officers face stigma within the community, and this is another reason to be careful with how their quotes are represented as to not further stigmatize them. Just something to be mindful of but I do understand the importance of those individuals' voices being represented within the study. 

Lastly, within the limitations section... 

LN 660-662 - I would suggest giving your study more credit here! It is, of course, important to acknowledge the lack of generalizability for the reasons you indicated but I would also suggest reiterating that your research group chose a qualitative approach for many reasons including that to gain a depth of understanding. You could also suggest future quantitative research efforts that could be conducted based on your findings for generalizability, etc.  

The other thing I would appreciate mentioned in the Limitations section is the age range of participants since everyone involved was 30 and over. It may be nice if you suggested any differences that you may expect or would be interested in hearing about if younger MHA nation community members would have participated particularly since many of the interventions suggested by participants were aimed at youth and family education / engagement.

Again, overall, this is a phenomenal paper! The knowledge from this study will hopefully be integrated into future courses on animal welfare at the undergraduate and veterinary level - I know it is something I will want to make known for my future students. Thank you for this great contribution to the scientific community and the amount of respect you provided to the indigenous community that you worked with.

Author Response

My concerns mainly focus on confidentiality. As this is a qualitative study, I find the number of interviews appropriate with the focus of gaining a depth of knowledge rather than generalizability. However, there were 14 participants. In my experience with qualitative and quantitative studies working with human participants, the general rule of thumb is to not identify a subgroup with 3 or less individuals as these participants could be re-identified. 

LN 129-132: I think it’s interesting to know the different roles of the participants within the tribal community. However, I am concerned that the phrasing of this sentence could re-identify individuals, specifically the animal control officers, journalist and school principal. I would suggest revising this to protect the individual participants identities.

RESPONSE: We sincerely appreciate your efforts to support us in protecting the confidentiality of the study participants! We removed some of the more specific titles in Lines 137 to address this concern.

I would also suggest removing Table 1 (I know, I usually love a demographics table - this is a weird suggestion from me!) but I think you could just briefly describe the participants breakdown for gender and age and include a statement that all participants self-identified as Native American, American Indian and/or First Nations. 

RESPONSE: Since removing the table doesn’t provide any more or less confidentiality for the subjects and providing a demographic table is standard practice for qualitative results, we’d prefer to keep the table, if the reviewers and editors are agreeable.

Methods question - Was any compensation or direct / indirect benefits provided to participants? I would suggest adding a statement that outlines that in the Methods section. 

RESPONSE: We added the compensation information to Lines 146-147

With regards to the Results and how statements were presented, I had a few more confidentiality concerns mostly related to how quotes were presented from participants that were identified by their role in the community, as it seems that they could be easily re-identified, and this could pose a risk to their reputation within their community. These quotes are all impactful and I believe that they should stay in the manuscript; however, I would suggest revising the presentation of the following lines to ensure that participant confidentiality is protected:

LN 266-269

RESPONSE: We have updated Lines 272-273, to use the more general “other community members” rather than the specific role of the participant. 

LN 287-291

RESPONSE: We also updated Lines 293-295 to us the more general “one participant.”

LN 314-317 * This may not need changed if you had > 3 animal control officers who participated.

RESPONSE: We did not make changes to Lines 319-321 since that descriptor was already using the general “another member.” 

LN 330-337 * This may be OK again if you had > 3 animal control officers who participated... However, may suggest removing the participant identifiers in general, ex. P7 as its clear P7 is an animal control officer.

RESPONSE: We also updated Lines 337-346 to us the more general “one participant” and made some minor edits to the quote for this participant to replace “we” with “animal control officers.”

LN 362-367

RESPONSE: We did not make changes to Lines 367-370 since that descriptor was already using the general “one interviewee.” 

LN 461-477 * NOTE: I see how impactful these statements are because they come from the veterinarian and the member who works for the MHA Nation's Fish and Wildlife Division so they have that professional role; however, this would make each of these individuals easily re-identified.

RESPONSE:  We did not make changes to Lines 464-469 since that descriptor was already using the general “another tribal member.” We did edit lines 470-484 to de-identify the veterinarian and Fish and Wildlife division individuals who participated. 

Also, I understand from the article that the animal control officers face stigma within the community, and this is another reason to be careful with how their quotes are represented as to not further stigmatize them. Just something to be mindful of but I do understand the importance of those individuals' voices being represented within the study. 

RESPONSE: We greatly appreciate your attention to this detail and believe that your suggested revisions will achieve both goals!

Lastly, within the limitations section... 

LN 660-662 - I would suggest giving your study more credit here! It is, of course, important to acknowledge the lack of generalizability for the reasons you indicated but I would also suggest reiterating that your research group chose a qualitative approach for many reasons including that to gain a depth of understanding. You could also suggest future quantitative research efforts that could be conducted based on your findings for generalizability, etc.  

RESPONSE: We greatly appreciate the suggestion here to highlight the merits of this qualitative approach and potential opportunities for quantitative research to build on these findings. We have added a brief discussion of this to the limitations section.

The other thing I would appreciate mentioned in the Limitations section is the age range of participants since everyone involved was 30 and over. It may be nice if you suggested any differences that you may expect or would be interested in hearing about if younger MHA nation community members would have participated particularly since many of the interventions suggested by participants were aimed at youth and family education / engagement.

RESPONSE: Thank you for this suggestion! We have added mention of the sample’s age bias and the suggested future direction to involve youth to the limitations section.

Again, overall, this is a phenomenal paper! The knowledge from this study will hopefully be integrated into future courses on animal welfare at the undergraduate and veterinary level - I know it is something I will want to make known for my future students. Thank you for this great contribution to the scientific community and the amount of respect you provided to the indigenous community that you worked with.

RESPONSE: Thank you for your careful review and enthusiasm for this manuscript!

Round 2

Reviewer 2 Report

Thanks to the authors for updating

I am now happy to go ahead with publishing

good luck